# Comparative Analysis of Circadian Transcriptomes Reveals Circadian Characteristics between *Arabidopsis* and Soybean

**DOI:** 10.3390/plants12193344

**Published:** 2023-09-22

**Authors:** Xingwei Wang, Yanfei Hu, Wei Wang

**Affiliations:** 1State Key Laboratory for Protein and Plant Gene Research, School of Life Sciences, Peking University, Beijing 100871, China; pkuwangxw@pku.edu.cn (X.W.); huyanfei2021@stu.pku.edu.cn (Y.H.); 2Center for Life Sciences, Beijing 100871, China

**Keywords:** circadian clock, *Arabidopsis*, soybean, transcriptome

## Abstract

The circadian clock, an endogenous timing system, exists in nearly all organisms on Earth. The plant circadian clock has been found to be intricately linked with various essential biological activities. Extensive studies of the plant circadian clock have yielded valuable applications. However, the distinctions of circadian clocks in two important plant species, *Arabidopsis thaliana* and *Glycine max* (soybean), remain largely unexplored. This study endeavors to address this gap by conducting a comprehensive comparison of the circadian transcriptome profiles of *Arabidopsis* and soybean to uncover their distinct circadian characteristics. Utilizing non-linear regression fitting (COS) integrated with weights, we identified circadian rhythmic genes within both organisms. Through an in-depth exploration of circadian parameters, we unveiled notable differences between *Arabidopsis* and soybean. Furthermore, our analysis of core circadian clock genes shed light on the distinctions in central oscillators between these two species. Additionally, we observed that the homologous genes of *Arabidopsis* circadian clock genes in soybean exert a significant influence on the regulation of flowering and maturity of soybean. This phenomenon appears to stem from shifts in circadian parameters within soybean genes. These findings highlight contrasting biological activities under circadian regulation in *Arabidopsis* and soybean. This study not only underscores the distinctive attributes of these species, but also offers valuable insights for further scrutiny into the soybean circadian clock and its potential applications.

## 1. Introduction

Residing on Earth, a multitude of organisms have developed a circadian clock system to synchronize their biological activities with the environment for their growth and well-being. These organisms span prokaryotes like cyanobacteria and eukaryotes such as yeast, plants, and animals. The circadian clock within them is pivotal for adaptation to adverse conditions. Particularly in plants, the circadian clock enables them to anticipate environmental shifts like temperature fluctuations and pathogen invasions [1,2,3]. By integrating external signals with internal cues, the plant circadian clock orchestrates diverse biological events, ensuring their occurrence at the optimal times throughout the day [4,5,6]. Given that the circadian clock governs the timing of biological activities throughout a plant’s life, understanding of its function has proven instrumental in enhancing plant vitality and prosperity.

At its core, the plant circadian system comprises central oscillators, input pathways, and output pathways [7,8]. Research on the circadian clock of the model plant *Arabidopsis* has predominantly focused on the central circadian oscillator and its interplay with input and output pathways [9]. Currently, over twenty clock or clock-associated components have been identified in *Arabidopsis* [7]. Within this circadian system, the central oscillator encompasses a series of interlocked transcription–translation feedback loops, while the input and output pathways facilitate the transmission of signals between external and internal environments to regulate various processes. Distinct clock proteins become active at specific times of the day, mutually influencing the expression of other clock genes [4,10]. The core interconnected feedback loop of the *Arabidopsis* circadian clock involves *CCA1/LHY* and *TOC1* [11,12,13,14,15]. *CCA1* and *LHY* peak at dawn, while *TOC1*’s expression peaks around dusk. *CCA1/LHY* and *TOC1* mutually suppress each other’s expression [16,17]. Following a reduction in CCA1/LHY protein levels post-dawn, the repression on *TOC1* subsides, resulting in increased TOC1 protein levels. With elevated TOC1 protein levels, *CCA1* and *LHY* expression are further restrained. The inverse process unfolds during the night. Some clock proteins can also repress their own expression when overexpressed. Furthermore, the clock system incorporates greater complexity, featuring multiple loops and numerous other proteins such as PRR5, ELF3, RVE1 etc. [18,19,20].

The *Arabidopsis* circadian clock has been extensively investigated. Likewise, research on the circadian clock of various crops has been expanding [21,22]. For instance, it has been reported that *OsCCA1* regulates ABA signaling pathways to enhance rice’s abiotic tolerance [23]. *TaELF3* can impact the heading dates of wheat through its effects on photoperiodic responses in circadian oscillators [24]. In maize, the evening complex of the circadian clock promotes flowering and adaptation to temperate regions [25]. Furthermore, studies have indicated tight correlations between circadian rhythms and abiotic stress in soybeans [26]. Despite these scattered documentations of the function of the circadian clock system in different crops, systematic comparative studies of circadian rhythms between *Arabidopsis* and crops remain limited. Given our scarce understanding of crop circadian clocks, such comparisons may produce valuable insights into the circadian clock’s role within crops.

The analysis of expressed genes through transcriptome profiling has proven integral in plant circadian clock research [27,28,29]. Specifically, time-course circadian transcriptome profiling offers a comprehensive approach for identifying a wide array of circadian rhythmic genes [26,30,31]. In this study, we undertook a comparative analysis of circadian transcriptome profiles in *Arabidopsis* and soybean to unveil their unique circadian attributes. Through the comparison of rhythmic genes and their corresponding circadian parameters, we discovered the distinctive characteristics of *Arabidopsis* and soybean. Additionally, an examination of core circadian clock genes identified inherent differences in phase, period, and amplitude between the two organisms. Finally, we unveiled that translation activities are more likely under circadian regulation in *Arabidopsis*, while photosynthesis activities are more prone to circadian control in soybean. These findings provide insights into the soybean circadian clock and establish a foundation for future engineering of the soybean circadian clock to improve the yield of soybean.

## 2. Results

### 2.1. Comparison of Circadian Rhythmic Genes Unveils Distinctive Circadian Parameter Characteristics between Arabidopsis and Soybean

We retrieved *Arabidopsis* and soybean circadian time-course transcriptome profiles from two independent studies which share consistent sampling schemes (Figure 1) [26]. In both studies, plants were initially cultivated in a light–dark cycle (LD) for a duration of 9 to 11 days. Subsequently, the plants were transferred to a continuous light condition (LL). Under the LL condition, timepoints were denoted using zeitgeber time (ZT), starting from ZT0. One day after transferring plants into the LL condition, samples of *Arabidopsis* and soybean were collected over the course of two consecutive days at 4 h intervals. These samples were then subjected to RNA-seq analysis, and the resulting raw sequencing reads were processed to generate the raw count matrix of gene expression data.

Following the pre-processing of the expression data, a total of 21,013 genes for *Arabidopsis* and 33,543 genes for soybean were retained and considered expressed genes. The good reproducibility of samples from the same timepoint is evident in the sample correlation heatmap (Appendix A). Notably, samples collected at the 24 h interval exhibit a higher correlation compared to those collected at the 12 h interval (Appendix A). Leveraging non-linear regression fitting (COS) integrated with weights analysis, oscillatory parameters were estimated for all expressed genes. This comprehensive dataset of expressed genes and their corresponding oscillatory parameters served as the foundation for subsequent analyses.

Prior to delving into the classification of circadian rhythmic genes, a thorough examination of the estimates of circadian oscillatory parameters was performed. Firstly, we defined circadian oscillation correlation as an index used to assess rhythm robustness and detect rhythmic genes. Circadian oscillation correlation refers to the correlation coefficient of observed data and predicted data from COS fitting. Higher circadian oscillation correlation indicates better oscillation of genes. The distribution of circadian oscillation correlations among the expressed genes demonstrates an apparent disparity between *Arabidopsis* and soybean (Figure 2a). Notably, for circadian oscillation correlation exceeding 0.8, *Arabidopsis* exhibits a higher density compared to soybean. Conversely, the absolute number of rhythmic genes in soybean consistently surpasses that of *Arabidopsis* across a range of correlation cutoffs (Figure 2b). This outcome is anticipated given the substantially larger number of soybean genes. However, when the proportion of rhythmic genes relative to the total expressed genes was considered, no statistically significant difference was observed between the two organisms (Figure 2c). This suggests that during the recent conversion from the tetraploid to the diploid genome [32], circadian rhythmic genes were not significantly retained or depleted in the soybean genome.

Previous studies have applied a circadian oscillation correlation of no less than 0.7 as the threshold in their research, and this has proven to be a robust cutoff [33,34]. Therefore, we adopted a circadian oscillation correlation of no less than 0.7 as a threshold to detect circadian rhythmic genes. This yielded a percentage of 44.98% and 42.51% of circadian rhythmic genes among the expressed genes of *Arabidopsis* and soybean, respectively. The yielded number of rhythmic genes are enough for down-stream analysis.

A comprehensive comparison of the circadian rhythmic genes of *Arabidopsis* and soybean was executed through an in-depth analysis of their oscillatory parameters. Phase24 is a period-corrected estimation of the peak expression time of circadian rhythmic genes [26]. The distribution of the phase24 of circadian rhythmic genes in *Arabidopsis* manifests a major peak at 10.97 h, with two minor peaks at 1.57 h and 22.60 h. However, the peak at 11.62 h is dramatically reduced, while the peak at 22.3 h becomes more dominant in soybean circadian rhythmic genes (Figure 3a). Consistently, a Kolmogorov–Smirnov test suggests statistically significant different distribution patterns of the phase24 of *Arabidopsis* and soybean circadian rhythmic genes (*p* < 0.001). The period distribution of soybean rhythmic genes is also different from that of *Arabidopsis*. Soybean rhythmic genes have significantly longer periods than those of *Arabidopsis* genes (*p* < 0.001, Mann–Whitney test, Figure 3b). Additionally, soybean rhythmic genes exhibit a higher amplitude than those of *Arabidopsis* (*p* < 0.001, Mann–Whitney test, Figure 3c). Notably, the average expression level of soybean rhythmic genes in is lower than that of *Arabidopsis* genes (*p* < 0.001, Mann–Whitney test, Figure 3d). These statistical analyses unequivocally indicate significant distinctions across all four key oscillation parameters. Importantly, these conclusions remain the same when alternative circadian oscillation correlation cutoffs, such as ≥0.5/0.6/0.8/0.9, are adopted (Appendix A). Collectively, these findings reveal that the *Arabidopsis* and soybean rhythmic genes display drastically different oscillatory characteristics.

### 2.2. Comparative Analysis of Homologous Circadian Clock Genes Implies Intricate Circadian Regulatory Mechanisms Distinguishing Soybean from Arabidopsis

To delve deeper into the variations in circadian regulation, we conducted an in-depth comparative analysis of the core clock genes in *Arabidopsis* and their corresponding homologous genes in soybean. The details of *Arabidopsis* gene names and IDs, alongside their counterparts in soybean, are outlined in Appendix A. Previous studies have indicated that *Arabidopsis CCA1* and *LHY* constitute partially redundant genes essential for circadian rhythm maintenance [14,35]. We identified their homologous genes in soybean. By comparing their oscillation parameters, we observed that no *GmCCA1/LHY* genes exhibit significantly changed periods compared to *CCA1* (Figure 4b). However, all *GmCCA1/LHY* genes, except *GmCCA1/LHY_5*, demonstrate lower amplitudes than *Arabidopsis CCA1* (Figure 4c). The oscillations of *CCA1*, *LHY*, and all *GmCCA1/LHY* genes are well pronounced (Figure 4a and Appendix A), with *GmCCA1/LHY_4* displaying an apparent phase shift (Figure 4a and Appendix A). These findings underscore diverse changes across various *GmCCA1/LHY* genes when compared with *CCA1*.

Given that *CCA1* and *LHY* are fundamental constituents of the morning loop within the central oscillators of *Arabidopsis*, while *TOC1* serves as the core component of the evening loop, we extended our comparison to *TOC1* and its homologous genes in soybean. Notably, there is no discernible difference in period between *TOC1* and its soybean-homologous genes (Figure 4e). While the amplitudes of *GmTOC1_1* and *GmTOC1_2* closely mirror that of *TOC1*, *GmTOC1_3* displays a lower amplitude, whereas *GmTOC1_4* exhibits a higher amplitude (Figure 4f). Generally, all *GmTOC1* genes exhibit better oscillations than *TOC1*, with a negligible disparity in phase (Appendix A).

Next, we directed our attention to the key clock genes within the *Arabidopsis* central oscillators (Figure 5). Among the analyzed soybean homologous genes corresponding to the 15 Arabidopsis clock genes, only *GmPRR3*, *GmZTL* and *GmELF3* exhibited a noteworthy alteration in period (Appendix A). Meanwhile, among the soybean homologous genes linked to 13 of the *Arabidopsis* clock genes, there was a notable finding: for each of these genes, at least one ortholog exhibited significant alterations in amplitude (Appendix A). With the exception of *GmPRR9*, the remaining soybean homologous genes demonstrated a significant phase shift (Appendix A). These shifts in the circadian expression patterns of circadian clock genes imply intricate and distinct regulatory mechanisms governing the central oscillators between *Arabidopsis* and soybean.

### 2.3. Circadian Parameter Alterations of Genes Provide Insights into the Circadian Control of Flowering and Maturity in Soybean

Numerous genetic loci, including those designated as the *E* series and *J* (long juvenility gene), have been cloned and determined to exert regulatory control over the flowering and maturation processes in soybean [36]. Remarkably, among these genes, *E2*, *E3*, *E4*, and *J* have been recognized as homologous counterparts of *Arabidopsis* clock genes [37,38,39,40].

Locus *J* is the ortholog of *Arabidopsis EARLY FLOWERING 3* (*ELF3*), and *J* can repress the transcription of *E1,* which is a legume-specific flowering repressor [41], so as to promote flowering under short days [40]. In this study, we found that the phase of *J* (*GmELF3_1*, GLYMA_04G050200) shifts from 15.87 (phase of *Arabidopsis ELF3*) to 14.11, and *J* (*GmELF3_1*) shows better oscillation than *ELF3* (Figure 6a and Appendix A). Moreover, the period of *J* shows no difference from that of *ELF3* (Appendix A). However, the amplitude of *J* (*GmELF3_1*) is significantly higher than that of *ELF3* (Appendix A). The four orthologs of *ELF3* show diverse circadian expression patterns (Figure 5n), implying potentially different functions.

Locus *E2* is the ortholog of *Arabidopsis GIGANTEA* (*GI*), and *E2* can repress flowering in soybean [38]. In this study, we found that the phase of *E2* (*GmGI_1*, GLYMA_10G221500) significantly shifts from 9.02 (phase of *Arabidopsis GI*) to 7.74, and *E2* shows better robustness of oscillation than *GI* (Figure 6b and Appendix A). However, the period of *E2* is no significantly different from *GI* (Appendix A). Additionally, the amplitude of *E2* (*GmGI_1*) is no different from *GI* (Appendix A).

Loci *E3* and *E4* are orthologs of *Arabidopsis phytochrome A* (*PHYA*) [37]. *E3* and *E4* could respond to different degrees of light under LDs and repress soybean flowering [42]. Details regarding the corresponding gene name and gene ID of *PHYA* and its orthologs in soybean are listed in Appendix A (under “PHYA” tab). *E3* (*GmPHYA_3*, GLYMA_19G224200) and *E4* (*GmPHYA_1*, GLYMA_20G090000) present obvious phase shifts compared with *PHYA* (Figure 6c). An interesting phenomenon is that *E4* (*GmPHYA_1*) oscillates better than *PHYA,* while *E3* (*GmPHYA_3*) oscillates much worse than *PHYA*. Besides, *E4* (*GmPHYA_1*) shows no difference in period compared to *PHYA* and has higher amplitude than *PHYA* (Appendix A). However, *E3* (*GmPHYA_3*) shows a shorter period and lower amplitude than *PHYA* (Appendix A).

These findings showcase a multitude of divergent traits between *Arabidopsis* clock genes and their corresponding orthologs implicated in the regulation of flowering and maturity within soybean. These results imply that the crucial roles played by circadian clock genes in governing the intricate processes of flowering and maturation may be the result of alterations of key circadian parameters.

### 2.4. Circadian Control of Physiological Activities in Arabidopsis and Soybean

While the abovementioned findings have allowed us to compare the attributes of central oscillators in *Arabidopsis* and soybean, the associated physiological activities regulated by these circadian clock genes remain to be investigated. To maximize the information derived from COS analysis, we conducted a gene set enrichment analysis (GSEA) utilizing the circadian oscillation correlation from both *Arabidopsis* and soybean. The enriched gene ontology (GO) terms were ranked by normalized enrichment score (NES). A higher NES implies a greater concentration of genes associated with the GO term at the top of the gene list, indicating a higher circadian oscillation correlation and better rhythmicity. We observed that both organisms feature enriched GO terms associated with rhythmic process-and photosynthesis-related functions (Figure 7). However, when it comes to molecular function (MF) and cellular component (CC) terms, *Arabidopsis* rhythmic genes are also enriched with many ribosomal-associated terms, while soybean rhythmic genes are still primarily enriched with GO terms associated with the photosystem (Figure 7). The expression profiles of rhythmic genes associated with the enriched GO term “cytosolic large ribosomal subunit” in *Arabidopsis* are presented in Appendix A, while those associated with the enriched GO term “photosystem” in soybean are profiled in Appendix A. This disparity suggests that rhythmic genes in *Arabidopsis* are more linked with translation processes in addition to photosynthesis, whereas their counterparts in soybean appear to correlate predominantly with photosynthesis.

### 2.5. Differential Circadian Regulation of Physiological Activities in Arabidopsis and Soybean

To enable a better dissection of the distinctive circadian regulations on physiological activities in *Arabidopsis* and soybean, we employed a tailored approach to filter GO terms. We compared the relationships among these GO terms and categorized overlapping terms into four groups based on their NES.

Initially, we employed FDR ≤ 0.05 as a threshold, yielding 581 enriched GO terms in *Arabidopsis* and 454 enriched GO terms in soybean. Of these, 194 terms were found to overlap between two organisms (Figure 8a). Notably, a significant positive correlation between the NESs derived from *Arabidopsis* and those from soybean is evident in these overlapping terms (*p* < 0.0001, *F* test, Figure 8b). This suggests that *Arabidopsis* and soybean share a consistent circadian regulation on the physiological activities associated with these terms.

To allow a more exhaustive survey of the distinctions of the circadian regulations between *Arabidopsis* and soybean, we reduced the threshold to *p*-value ≤ 0.05, since the multiple comparison corrections used to derive FDR are not strictly required due to the inherent algorithm difference between GSEA and a traditional GO enrichment analysis [43]. This resulted in 1377 GO terms for *Arabidopsis* and 1067 GO terms for soybean. Among these, 444 terms are shared by both organisms (Figure 8c). Intriguingly, eight GO terms were classified in the upper-left region of the NES diagram (Figure 8d), indicating positive *Arabidopsis* NESs but negative soybean NESs. These eight terms are predominantly associated with translation, as outlined in more details in Table 1. This indicates that translation-related processes are enriched with circadian rhythmic genes in *Arabidopsis* but depleted in soybean, suggesting differential involvement of circadian regulations on translation in these two organisms.

To further reveal the distinctions in the circadian regulation of physiological activities between these *Arabidopsis* and soybean, we carried out an enrichment map analysis targeting the BP terms of the 387 GO terms uniquely enriched in *Arabidopsis*, as well as the BP terms of the 260 GO terms specifically enriched in soybean (GO terms from Figure 8a). The enrichment map pertaining to *Arabidopsis* unveiled an apparent concentration of pathways related to translation, response to stimuli and nucleobase biosynthetic processes (Figure 8e, and the whole map elaborated in Appendix A). On the other hand, the enrichment map of soybean identifies enrichment of activities associated with the regulation of nuclear division and cysteine metabolic processes (Figure 8f, with the whole map information in Appendix A).

Finally, we looked into all the 6462 GO terms from *Arabidopsis* and 5135 GO terms from soybean without any specific cutoffs to allow the most relaxed and inclusive comparison. Among these, 4603 terms are shared by *Arabidopsis* and soybean (Appendix A). These terms can be separated into four regions in the NES diagram (Appendix A).

Within each of these regions, we systematically sorted the terms based on the ascending order of *p* values. Subsequently, we extracted the top five terms from the upper-left region indicating positive *Arabidopsis* NESs but negative soybean NESs (Figure 9a–e). Impressively, the predominant theme of these five terms is associated with ribosome and translation processes. This compelling observation suggests a heightened likelihood of translation activities being regulated by the circadian clock in *Arabidopsis*, whereas *Glycine max* might deviate from circadian regulation in this context. In parallel, we identified the top five terms from the lower-right region signifying positive soybean NESs but negative *Arabidopsis* NESs (Figure 9f–j). Interestingly, these five terms are notably related to phosphogluconate dehydrogenase (decarboxylating) activity and related functions. This pattern implies that specific enzyme activities in soybean are more prone to circadian regulation, potentially differing from the norm in *Arabidopsis*. Further specifics about these ten terms are outlined in Table 2.

## 3. Discussion

The circadian clock is a fundamental timekeeping mechanism that allows organisms to synchronize their biological activities with the external environment, ensuring optimal growth, development, and adaptation. *Arabidopsis* is an important model organism extensively studied to dissect the intricate molecular underpinnings of circadian rhythms. *Glycine max*, also known as soybean, is a major crop utilized in many aspects of human life. This study employed a comprehensive comparative analysis of circadian time-course transcriptome profiles to uncover the circadian characteristics that differentiate these two species, shedding light on the investigations and applications of the intricate regulatory mechanisms in soybean’s circadian clock.

Firstly, with a circadian oscillation correlation no less than 0.7 as a threshold, we obtained a percentage of 44.98% and 42.51% of circadian rhythmic genes among the expressed genes of *Arabidopsis* and soybean, respectively. The percentage in *Arabidopsis* is a little higher than that in soybean. However, both percentages are in common regions, as reported in previous study which observed that about 5.2% to 55.9% of genes show significantly rhythmic expression across species in Archaeplastida [44]. Besides, some studies have reported that about 50% of genes are rhythmic in mammalian animals [45,46]. Therefore, the percentage of rhythmic genes in *Arabidopsis* and soybean are somehow consistent with that in broad species.

Then, with the analysis of circadian rhythmic genes, there were significant variations in the expression patterns, phase24, period, and amplitude of these genes. Soybean homologous genes of *Arabidopsis* rhythmic genes were different in their expression patterns and generated distinct rhythmicity. For example, *Arabidopsis PHYA* (circadian oscillation correlation = 0.97) and soybean *GmPHYA_1* (circadian oscillation correlation = 0.96) present good rhythmicity, while soybean *GmPHYA_4* (circadian oscillation correlation = 0.59) is not rhythmic. Besides, our analysis also showed the differentiation of homologous clock genes in phase24, period, and amplitude. These differences underscore the intricate nature of the circadian regulatory mechanisms within each species. More comprehensive details of the differentiation phenomenon remain a concern for future investigation.

Further analysis of the circadian clock genes provided deeper insights into the circadian control of flowering and maturity. Homologous genes like *GmGI* and *GmELF3* of *Arabidopsis* clock genes in soybean were previously found to play pivotal roles in the regulation of these critical developmental processes [38,40]. The circadian parameter alterations of these homologous genes in soybean emphasize the importance of circadian genes in the regulatory pathways across different species. In addition, previous studies have reported that the period and phase shifts of clock genes were relevant to the domestication of cultivated tomato [47]. It cannot be denied that other homologous genes of *Arabidopsis* clock genes in soybean may also play roles in a wide range of applications.

Delving into the enrichment map analysis, this study uncovered the specific biological processes subjected to circadian regulation in each organism. Interestingly, while some processes showed overlapping patterns of regulation, such as rhythmic processes and photosynthesis-related terms, others exhibited distinct patterns. The variations in circadian regulation were evident in processes associated with translation, response to stimuli, and nucleobase biosynthetic processes in *Arabidopsis*, while in soybean, activities tied to regulation of nuclear division and cysteine metabolic processes stood out. These differential regulatory patterns likely reflect the main roles of rhythmic genes in each species.

A recent study identified starch metabolism as a clock-controlled pathway in hexaploid bread wheat, and provided important targets for future wheat breeding [31]. In this study, one of the remarkable findings was the divergent behavior of translation activities between *Arabidopsis* and soybean. While translation activities in *Arabidopsis* were more likely to be regulated by the circadian clock, circadian rhythmic genes were actually depleted in translation-related pathways in soybean. Another finding was that some activities, especially enzyme-associated activities such as phosphogluconate dehydrogenase (decarboxylating) activity, are more likely under circadian regulation in soybean. By separating common GSEA GO terms between *Arabidopsis* and soybean into four groups (Figure 8a–d and Appendix A), we identified terms showing opposite NES. Larger positive NES means the genes behind the term tend to concentrate in the top region of the whole gene list and tend to be more rhythmic. On the one hand, the results confirmed that translation activities in *Arabidopsis* were more likely to be regulated by the circadian clock (Table 1 and Table 2). On the other hand, the results indicated some activities, especially enzyme-associated activities, are more prone to circadian regulation in soybean (Table 2). This observation underscores the specific adaptability of circadian regulation in soybean and provides more directions for research about soybean circadian rhythms.

In conclusion, this comprehensive comparative analysis of circadian time-course transcriptome profiles provides a multifaceted understanding of the circadian characteristics that distinguish soybean from *Arabidopsis*. The findings extend insights into rhythmic genes and highlight the variations of biological processes between *Arabidopsis* and soybean, implying the complexity between circadian rhythms and biological activities. As our understanding of circadian regulation deepens, insights from this study may pave the way for targeted interventions in crop development, enhancing agricultural productivity and sustainability in a changing world.

## 4. Materials and Methods

### 4.1. Plant Materials and Growth Conditions

The *Arabidopsis* plants used in the study were wild-type, and of the Columbia (Col-0) ecotype. Col-0 seeds were sown on MS medium plates and cold-treated (4 °C, darkness) for 3 days. The MS plates were transferred to 22 °C under LD conditions (12 h light/12 h dark cycles; 50 μmol⋅m^−2^⋅s^−1^ of white light) for 11 days, and then placed under LL conditions (24 h light; 50 μmol⋅m^−2^⋅s^−1^ of white light).

The soybean seedlings used in the study were wild-type, and of the soybean cultivar Williams 82. Williams 82 seedlings were grown in soil under LD conditions (16 h light/8 h dark, 100 μmol⋅m^−2^⋅s^−1^, 28 °C, 50% relative humidity) for 9 days. On the tenth day, the light was switched to LL (24 h light; 100 μmol⋅m^−2^⋅s^−1^).

After a one-day transition under LL conditions, samples were harvested at 4 h intervals for 2 continuous days starting at ZT24. Samples of a total of 12 timepoints were collected. For *Arabidopsis*, each timepoint contains 2 biological replicates. For soybean, each timepoint contains 3 biological replicates. Samples were subjected to RNA extraction and RNA-seq. More details of *Arabidopsis* [30] and soybean [26] can be inferred from corresponding studies.

### 4.2. Sequencing Reads Acquisition, Read Alignment, and mRNA Quantification

Raw RNA-seq reads generated from *Arabidopsis* and soybean in this study can be retrieved from EMBL-EBI’s ArrayExpress (accession no. E-MTAB-7933) [30] and National Center for Biotechnology Information’s Gene Expression Omnibus (accession no. GSE94228) [26], respectively.

First, raw sequencing reads were analyzed using fastp [48] to remove low-quality reads and trim adapter sequences. Then, the qualified reads were aligned to the reference genome by TopHat2 (v2.1.1) [49] and assembled by StringTie (v2.1.5) [50] to obtain the quantified raw count matrix of gene expression.

### 4.3. Pre-Processing of Expression Data

The R packages limma [51] and edgeR [52] were used to process the raw count matrix of gene expression. First, function filterByExpr was selected to retain expressed genes which have sufficient large counts for downstream analysis. Then, scaling factors were calculated to normalize the raw library size for each sample. At the same time, the correlation matrix was calculated to check the reproducibility of samples. Lastly, the function voomWithQualityWeights was used to calculate weights for each sample to reduce the effect of variable samples on downstream statistical analysis. At the same time, normalized expression levels of genes were calculated as log_2_(CPM), derived from the voomWithQualityWeights function.

### 4.4. Weighted Estimation of Rhythmic Parameters of Genes

After pre-processing of the expression data, non-linear regression fitting (COS) integrated with weights was applied to each gene. The following formula was used in fitting.
(1)$$Expression\;level=Amplitude\times cos(\frac{2\pi }{Period}\times time-\frac{2\pi }{24}\times Phase24)+Constant$$

Each gene has 12 timepoints. For each gene, the rhythmic parameters were estimated using the fitting formula with sampling time as the independent variable and the corresponding sample’s expression level as the dependent variable. The fitting was carried out in the following way:(1)Remove the slope trends by fitting a linear regression model to the gene expression data and keep residuals.(2)Perform Fast Fourier Transformation (FFT) of the time series data and keep main signals.(3)Estimate the initial period using FFT transformed data.(4)Perform non-linear regression fitting. The fitting was applied with weights obtained in the pre-processing. Amplitude was constrained to be non-negative. Period was constrained to be greater than 12 h but less than 36 h. Phase24 and Constant have no constraints. Phase was normalized as phase24, which is more than 0 but less than 24. Constant indicates the average expression level of the gene.(5)Predict the best-fit data and calculate circadian oscillation correlation between observed data and predicted data.(6)Genes without convergent fits were considered arrhythmic. Genes with the resulting best-fit Amplitude, Period, Phase24, Constant and their standard error and degree of freedom were used for downstream statistical analysis.

### 4.5. Analysis of Homologous Genes

Homologs of *Arabidopsis* circadian clock genes in soybean were identified according to the method described previously [26]. Time-course expression levels of the *Arabidopsis* circadian clock gene and its homologous genes in soybean were profiled together using normalized data from the pre-processing. The rhythmic parameters of homologous genes in soybean were statistically tested with those of *Arabidopsis* circadian clock genes.

### 4.6. Phase24 Plots

Phase24 plots were shown as radial plot indicating differences in phase24 and oscillation robustness between *Arabidopsis* genes and their homologous genes in soybean. Phase24 indicates the gene’s phase normalized into a period of 24 h, and is plotted as the angular coordinate. Robustness is indicated by −log_10_(*p*) with a larger −log_10_(*p*) representing better oscillation. Lines with arrows indicate the phase24 shift and robustness change from the *Arabidopsis* genes to their homologous genes in soybean.

### 4.7. Gene Set Enrichment Analysis

Gene set enrichment analysis (GSEA) is a method to determine whether a group of specific genes tends to occur toward the top or bottom of the ranked whole gene list. In this study, GSEAs were performed using the R package clusterProfiler [53]. For *Arabidopsis* GSEA, the database org.At.tair.db was used. For soybean GSEA, database AH85411 from package AnnotationHub was used. The gene list was generated by sorting genes with their circadian oscillation correlation in descending order for *Arabidopsis* and soybean, respectively. Larger circadian oscillation correlation indicates better oscillation. The maximum size of gene set for analysis was set as 800. The minimum size of gene set for analysis was set as 3. Enriched GO terms including BP, MF, and CC were analyzed using the function gseGO in clusterProfiler. The network showing relationships and significance of BP terms was generated by the emapplot function in the package enrichplot [54]. The GSEA enrichment plots were generated using the gseaplot2 function in the package enrichplot. 

### 4.8. Statistical Analysis

The statistical methods and details are indicated in the methods or figure legends. Statistical tests and analysis were performed using RStudio software version 4.0.4.

## Figures and Tables

**Figure 1 plants-12-03344-f001:**
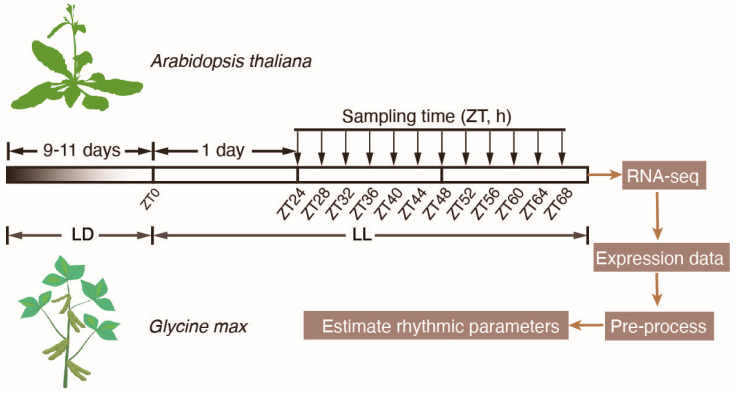
Sampling scheme of the two circadian time-course RNA-seq experiments. *Arabidopsis thaliana* and soybean seedlings were cultured under LD conditions for 11 and 9 days, respectively. After transfer to LL for 1 day, samples were harvested at 4 h intervals for 2 days. Arrows indicate the sampling timepoints.

**Figure 2 plants-12-03344-f002:**
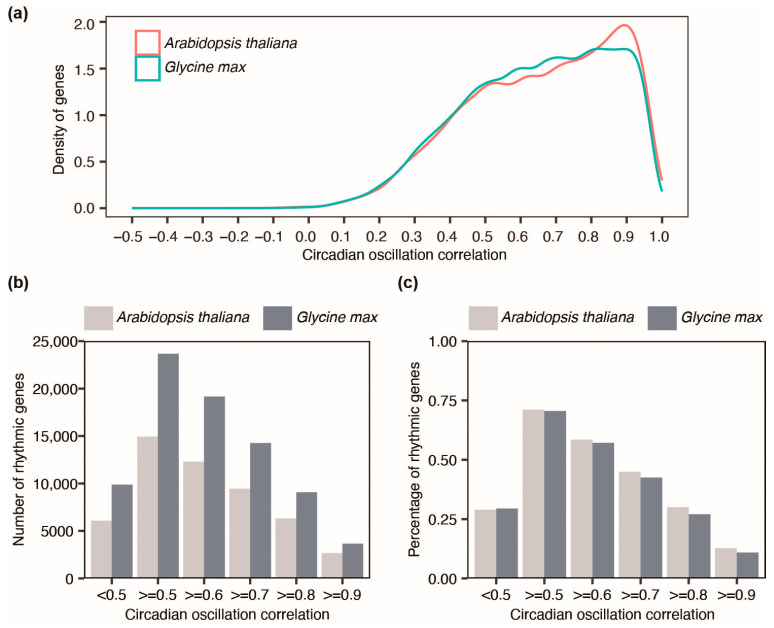
Exploratory analysis of circadian oscillation correlation of expressed genes in *Arabidopsis* and soybean. (**a**) Distribution of circadian oscillation correlations of expressed genes in the harvested samples under the experimental condition in *Arabidopsis* and soybean, respectively; (**b**) Number of rhythmic genes using different circadian oscillation correlation cutoffs in *Arabidopsis* and soybean, respectively; (**c**) Percentage of rhythmic genes using different circadian oscillation correlation cutoffs in *Arabidopsis* and soybean, respectively, Kolmogorov–Smirnov test *p* > 0.1. Note: there are a total of 21,013 and 33,543 genes expressed in the harvested samples of *Arabidopsis* and soybean under experimental conditions, respectively. The time-course expression profiles of genes were used for the estimation of circadian oscillatory parameters.

**Figure 3 plants-12-03344-f003:**
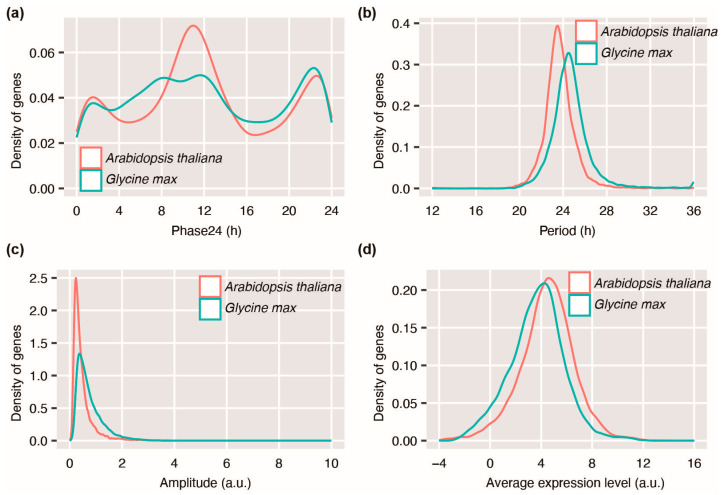
Global evaluation of oscillation parameters of circadian rhythmic genes with circadian oscillation correlation ≥ 0.7 as the cutoff. (**a**) Distribution of phase24 of rhythmic genes in *Arabidopsis* and soybean, Kolmogorov–Smirnov test *p* < 0.001; (**b**) Distribution of period of rhythmic genes in *Arabidopsis* and soybean, Mann–Whitney test *p* < 0.001; (**c**) Distribution of amplitude of rhythmic genes in *Arabidopsis* and soybean, Mann–Whitney test *p* < 0.001; (**d**) Distribution of average expression level of rhythmic genes in *Arabidopsis* and soybean, Mann–Whitney test *p* < 0.001. h, hour. a.u., arbitrary unit.

**Figure 4 plants-12-03344-f004:**
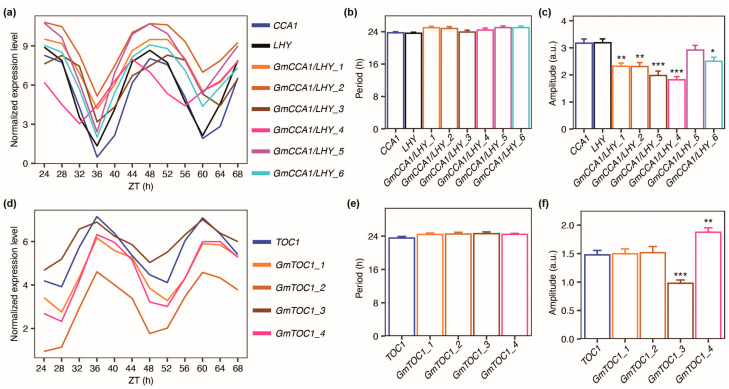
Comparison between *Arabidopsis* core loop clock genes *CCA1*/*LHY*, *TOC1* and their homologs in soybean. (**a**) Expression profiles of *Arabidopsis* genes *CCA1*/*LHY* and their homologous genes (from number 1 to 6) in soybean; (**b**) Period of *Arabidopsis CCA1*/*LHY* and their homologous genes in soybean; (**c**) Amplitude of *Arabidopsis CCA1/LHY* and their homologous genes in soybean; (**d**) Expression profiles of *Arabidopsis* gene *TOC1* and its homologous genes (from number 1 to 4) in soybean; (**e**) Period of *Arabidopsis TOC1* and its homologous genes in soybean; (**f**) Amplitude of *Arabidopsis* genes *TOC1* and its homologous genes in soybean. Normalized expression level refers to log_2_(CPM). In the bar graphs, data are presented as mean + SEM, * indicates *p* value < 0.05, ** indicates *p* value < 0.01, and *** indicates *p* value < 0.001 (one-way ANOVA followed by Holm–Šídák’s multiple comparisons test). Error bar, SEM.

**Figure 5 plants-12-03344-f005:**
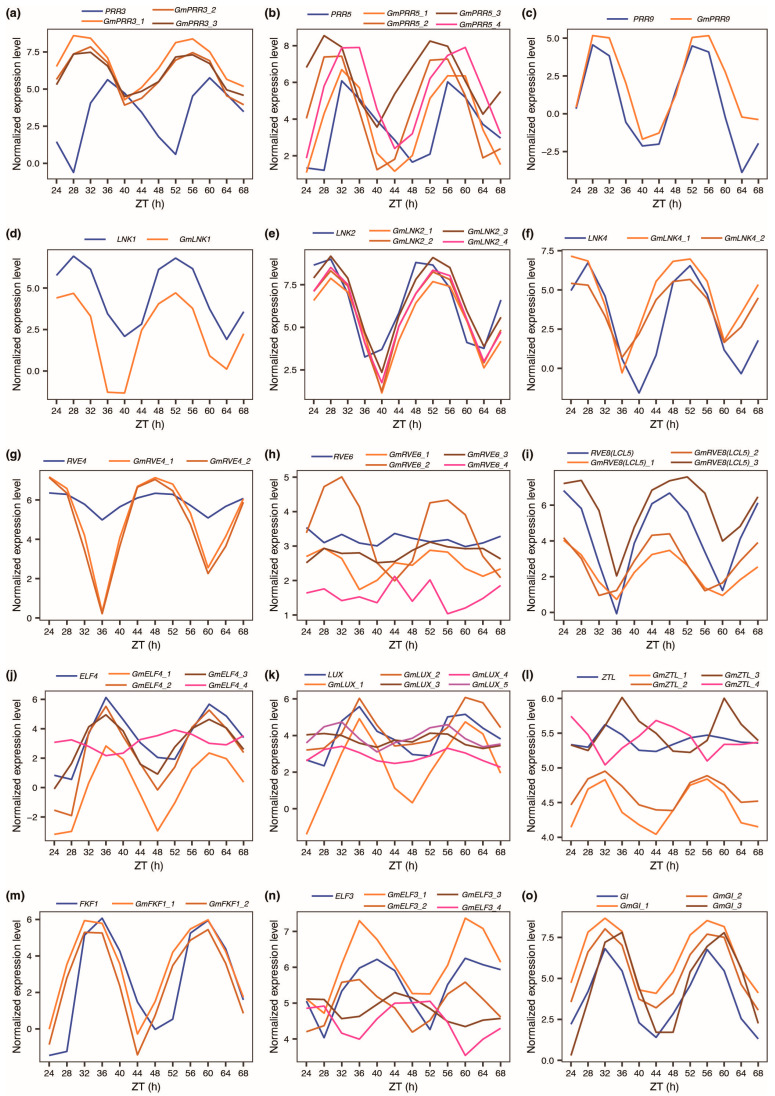
Expression profiles of *Arabidopsis* circadian clock genes and their homologous genes in soybean. (**a**) *Arabidopsis PRR3* and its homologous genes in soybean; (**b**) *Arabidopsis PRR5* and its homologous genes in soybean; (**c**) *Arabidopsis PRR9* and its homologous genes in soybean; (**d**) *Arabidopsis LNK1* and its homologous genes in soybean; (**e**) *Arabidopsis LNK2* and its homologous genes in soybean; (**f**) *Arabidopsis LNK4* and its homologous genes in soybean; (**g**) *Arabidopsis RVE4* and its homologous genes in soybean; (**h**) *Arabidopsis RVE6* and its homologous genes in soybean; (**i**) *Arabidopsis RVE8* and its homologous genes in soybean; (**j**) *Arabidopsis ELF4* and its homologous genes in soybean; (**k**) *Arabidopsis LUX* and its homologous genes in soybean; (**l**) *Arabidopsis ZTL* and its homologous genes in soybean; (**m**) *Arabidopsis FKF1* and its homologous genes in soybean; (**n**) *Arabidopsis ELF3* and its homologous genes in soybean; (**o**) *Arabidopsis GI* and its homologous genes in soybean.

**Figure 6 plants-12-03344-f006:**
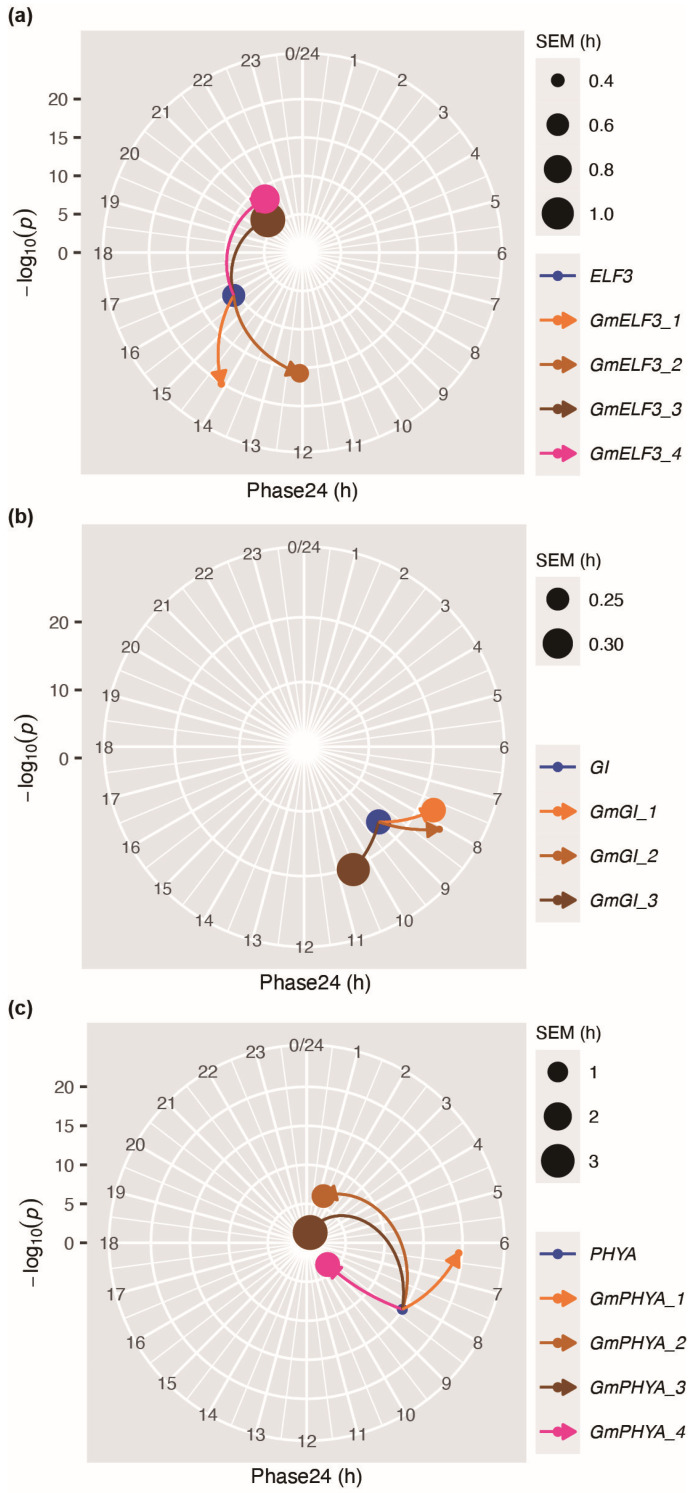
Radial plot showing differences in phase24 and oscillation robustness between *Arabidopsis ELF3* (**a**), *GI* (**b**), *PHYA* (**c**) and their homologous genes in soybean. Phase24 indicates a gene’s phase normalized into a period of 24 h, and is plotted as the angular coordinate. Robustness is indicated by −log_10_(*p*), with a larger −log_10_(*p*) value representing better oscillation. SEMs are indicated by the size of the symbols.

**Figure 7 plants-12-03344-f007:**
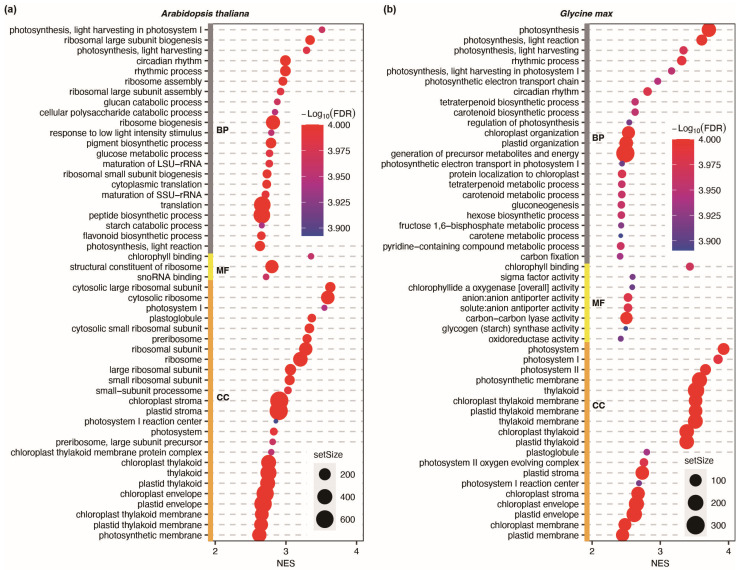
Top 50 enriched gene ontology terms sorted by NES from gene set enrichment analyses of *Arabidopsis* (**a**) and soybean (**b**), respectively. Genes are sorted in descending order by circadian oscillation correlation obtained from COS. NES, normalized enrichment score. BP, biological processes. MF, molecular function. CC, cellular components. FDR, false discovery rate.

**Figure 8 plants-12-03344-f008:**
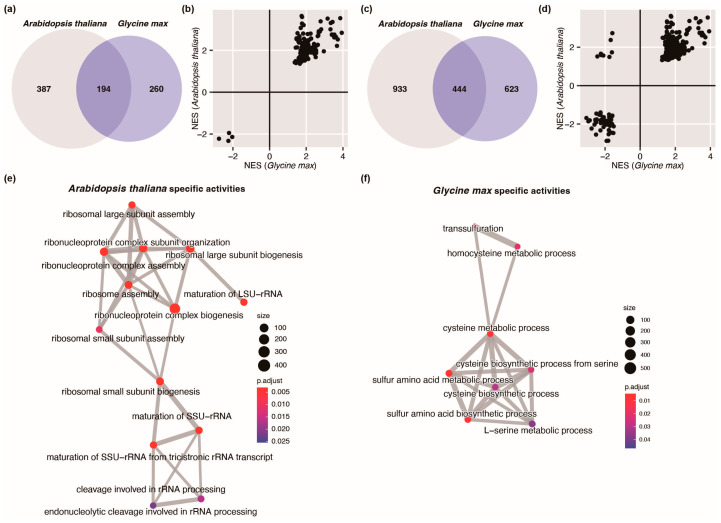
Comparison of GSEA results between *Arabidopsis* and soybean. (**a**) Venn diagram showing overlapped GO terms between *Arabidopsis* and soybean, derived via GSEA, with FDR ≤ 0.05 as a cutoff; (**b**) Comparison of the NESs of the 194 common GO terms from (**a**) between *Arabidopsis* and soybean; (**c**) Venn diagram showing overlapped GO terms between *Arabidopsis* and soybean derived from GSEA with *p* ≤ 0.05 as cutoff; (**d**) Comparison of the NESs of the 444 common GO terms from (**c**) between *Arabidopsis* and soybean; (**e**) Part of the relationship map of BP terms from 387 GO terms specifically enriched in *Arabidopsis* from (**a**); (**f**) Part of the relationship map of BP terms from 260 GO terms specifically enriched in soybean (**a**)*. p*.adjust indicates FDR. NES indicates the normalized enrichment score.

**Figure 9 plants-12-03344-f009:**
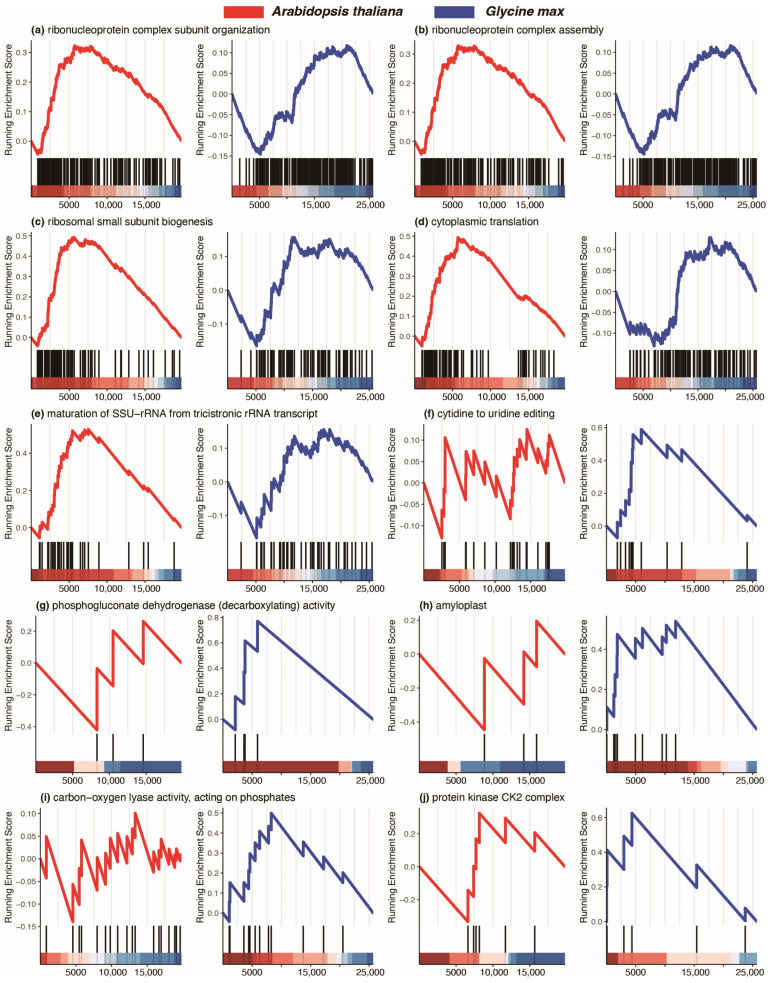
Comparison of GO terms between *Arabidopsis* and soybean with opposite NESs. Ten GO terms are plotted as (**a**–**j**). Each panel indicates a comparison of a GO term between *Arabidopsis* and soybean. Normalized enrichment scores (NES) and other details are listed in Table 1.

**Table 1 plants-12-03344-t001:** 8 GO terms from 444 overlapped terms between *Arabidopsis thaliana* and *Glycine max*.

			*Arabidopsis thaliana*			*Glycine max*		
Type	GO ID	Description	NES ^1^	*p* Value	Rank ^2^	NES	*p* Value	Rank
BP	GO:0042274	ribosomal small subunit biogenesis	2.7329	0.0001	5671	−1.5852	0.0405	20,587
CC	GO:0005852	eukaryotic translation initiation factor 3 complex	2.3767	0.0001	5596	−1.6797	0.0131	15,219
BP	GO:0055075	potassium ion homeostasis	1.7333	0.0126	9684	−1.6109	0.0496	8780
CC	GO:0000502	proteasome complex	1.5905	0.0146	10,956	−2.2152	0.0120	16,030
CC	GO:0005839	proteasome core complex	1.6584	0.0166	11,279	−2.2368	0.0013	16,579
CC	GO:1905369	endopeptidase complex	1.5202	0.0229	10,956	−2.4388	0.0169	16,030
BP	GO:0010499	proteasomal ubiquitin-independent protein catabolic process	1.5683	0.0364	11,279	−2.0977	0.0033	16,579
CC	GO:0070993	translation preinitiation complex	1.4947	0.0488	5486	−1.7524	0.0045	15,219

^1^, NES, normalized enrichment score. ^2^, Rank, order of the gene which corresponds to the enrichment score in the descending list.

**Table 2 plants-12-03344-t002:** Comparison of 10 GO terms from GSEA results between *Arabidopsis thaliana* and *Glycine max*.

	*Arabidopsis thaliana*				*Glycine max*			
Term	NES ^1^	*p* Value	FDR ^2^	Rank ^3^	NES	*p* Value	FDR	Rank
ribonucleoprotein complex subunit organization	1.965	0.000	0.003	5744	−1.933	0.333	0.674	20,531
ribonucleoprotein complex assembly	1.995	0.000	0.003	5744	−1.943	0.333	0.674	20,531
ribosomal small subunit biogenesis	2.745	0.000	0.003	5671	−1.633	0.037	0.199	20,587
cytoplasmic translation	2.741	0.000	0.003	5696	−1.311	0.185	0.494	18,680
maturation of SSU-rRNA from tricistronic rRNA transcript	2.491	0.000	0.003	7533	−1.331	0.083	0.323	20,587
cytidine to uridine editing	−0.743	0.842	0.944	17,268	2.032	0.001	0.022	5953
phosphogluconate dehydrogenase (decarboxylating) activity	−0.882	0.598	0.819	11,493	1.780	0.008	0.075	5914
amyloplast	−0.939	0.520	0.769	10,958	1.752	0.013	0.104	11,820
carbon-oxygen lyase activity, acting on phosphates	−0.732	0.847	0.948	15,240	1.721	0.015	0.112	8267
protein kinase CK2 complex	−1.005	0.422	0.703	13,202	1.684	0.019	0.131	4358

^1^, NES, normalized enrichment score. ^2^, FDR, false discovery rate. ^3^, Rank, order of the gene which corresponds to the enrichment score in the descending list.

## Data Availability

The raw RNA-seq reads generated from *Arabidopsis* and soybean in this study can be retrieved from EMBL-EBI’s ArrayExpress (accession no. E-MTAB-7933) [30] and the National Center for Biotechnology Information’s Gene Expression Omnibus (accession no. GSE94228) [26], respectively.

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
