# Peer review of "Comparative Analysis of Circadian Transcriptomes Reveals Circadian Characteristics between Arabidopsis and Soybean"

_plants, 2023, doi:10.3390/plants12193344_

Round 1

Reviewer 1 Report

The article is devoted to the actual topic of understanding plant circadian clock.
A detailed meta-analysis of two series of transcriptomic experiments for soybean and Arabidopsis is presented. The authors present the results of a detailed study of biological processes that are associated with circadian genes. The article is written in good language and provided with detailed illustrations.
As a side note, I think more links to other articles could be added to the discussion section. The authors discuss the results of their analysis in great detail, but provide few references to other articles on the topic of circadian rhythm.

The quality of the English language meets the requirements for scientific publications.

Author Response

Dear Professor,

I want to extend my sincere gratitude for your invaluable time and effort dedicated to reviewing our manuscript. Your commits are greatly appreciated. Discussion has been modified to link more references and emphasize the key points of this study. Please see the revised manuscript.

Best wishes,

Wei Wang

Reviewer 2 Report

In this manuscript by Wang er al., they systematically compared the transcriptomes between Arabidopsis and soybean. Their data suggest that translation activities is preferably under control by circadian clock, while the photosynthesis activity is more like control by clock in soybean. Moreover, the period distribution of rhythmic genes is different in soybean and in Arabidopsis, with soybean rhythimic genes have longer period, but lesser expression level. It is interesting to compare the expression patterns of clock core components between Arabidopsis and soybean. Overall, this manuscript is very informative, and well written. Nonetheless, It is unclear that why there are five copies of TOC1 homologue in soybean. Are they true homologs of TOC1 or as PRR family member’s homolog? Similarly, the four homologues of CCA1 and LHY in soybean have been reported, why there are 6 members in here?

Author Response

Dear professor,

Thank you for taking time to review this manuscript. We sincerely appreciate your comments. The responses are shown in the attachment.

Best wishes,

Wei Wang

Reviewer 3 Report

The manuscript is trying to comparative transcriptome between Arabidopsis and soybean focusing on the circadian regulation. The idea is interesting. However, I feel that the survey processes and discussions are not matured for publishing research articles.

First of all, the authors said that “Utilizing weighted FFT-NLLS analysis, we identified 15 circadian rhythmic genes within both organisms” in Abstract. However, for me, the fitting method described in 4.4. is seemed not to be FFT-NLLS, but a simple cosine fit. In the analysis with FFT-NLLS, it is common way to use RAE (Relative Amplitude Error) for discuss about rhythm significance. In this manuscript, instead of RAE, the authors used “circadian oscillation correlation” for exploring circadian rhythmic genes. I do not know the details of this index. And unfortunately, I could not find the definition of this index in the manuscript. The rhythmic gene detection is a critical basis for the comparative analysis. Please make sure the analysis process.

The fitting algorithm including FFT-NLLS may be suitable for long-term (> 3 days) measurement. For short-term measurement like this study (2 days), statistics algorithm such as MetaCycle may be suitable. To validate the result of FFT-NLLS by comparing with that of MetaCycle will be contribute to more acute comparison.

In the manuscript, the authors pointed out the differences between Arabidopsis and soybean. These two species are phylogenetically distant, so there should be some differences. In the discussion section, two paragraphs ended with “reflect the unique ecological niches and environmental adaptations of each species.” and “prioritize specific processes based on their ecological and physiological requirements.” But there is no hypothesis or consideration about which environmental differences are related to the observed differences. So, what is the new findings of this study? Please provide hypotheses. This manuscript is too descriptive to call “research paper” for me.

Author Response

Dear Professor,

I want to extend my sincere gratitude for your invaluable time and effort dedicated to reviewing our manuscript. Your commits are greatly appreciated. Please see the attachment for response details.

Best wishes,

Wei Wang

Reviewer 4 Report

COMMENTS TO THE AUTHOR:

Comments and Suggestions for Authors

This manuscript explained the distinctions of circadian clocks in Arabidopsis and soybean by conducting a comprehensive comparison analysis. The authors found that the circadian clock homologous genes exhibit different biological activities between Arabidopsis and soybean. This study shows well-planned experimental results, and these results are thought to be of interest to readers. I believe the manuscript is suitable for publication in the Plants after minor revisions.

Minor Concerns:

1. Line 110 – What is circadian oscillation correlation?

2. Line 121 – the authors give the threshold of circadian oscillation correlation no less than 0.7, an explanation should be given here.

3. in figure 4, what is the basis calculation of normalized expression level?

4. in Discussion, the distinction of expression, phase24, GO and GESA of circadian clock genes should be fully discussed and explored.

5. in MM 4.1, give the technique of seedlings growing: seed sterilization, conditions of germination and growing (temperature, light day/night, humidity).

6. in MM4.3, give the presentation of expression level, TPM, RPKM or FPKM.

7. The reference format requires a lot of checking.

The English is well written.

Author Response

Dear Professor,

The attachment is the newest response document. Thank you for your time.

Best wishes,

Wei Wang

Round 2

Reviewer 3 Report

Thank you for your kind responses.

Method explanation was really improved. I have no additional commments.

I have one comment for discuccsion. 

P.16 L414 ~ "Another finding was that photosystem appeared to be the clock-controlled components in soybean. This observation under scores the specific adaptability of circadian regulation in soybean and provides more directions for researches about soybean circadian rhythms. "

The enrichment of "photosystem" in GO term is not soybean specific according to Fig. 7. The circadian regulation of photosystem is generally considered  common in green linckages. Authors should explain more details of the comparision with A. thaliana to suggest that "This observation under scores the specific adaptability of circadian regulation in soybean".

Author Response

Dear Professor,

Thank you for your time and responsible efforts to review our manuscript. We really appreciate your comments which point out our omission of the term “photosystem” existing in Figure 7a and the incompatible conclusion. We have corrected our expression in the revised manuscript, line 414 to line 423. Specifically, the finding was that some activities especially enzyme-associated activities such as phosphogluconate dehydrogenase (decarboxylating) activity are more likely under circadian regulation in soybean. By separating common GSEA GO terms between Arabidopsis and soybean into 4 groups (Figure 8a-d, Figure S14), we identified terms showing opposite NES. Larger positive NES means the genes behind the term tend to concentrate in the top region of the whole gene list and tend to be more rhythmic. On the one hand, the results confirmed that translation activities in Arabidopsis were more likely to be regulated by the circadian clock (table 1-2). On the other hand, the results indicated some activities especially enzyme-associated activities are more prone to circadian regulation in soybean (table 2).

We hope this can answer your comments and make our manuscript better for publication in Plants.

Best wishes,

Wei Wang